# Neighbourhood Consensus Networks

**Ignacio Rocco**[†]  **Mircea Cimpoi**[‡]  **Relja Arandjelović**[§]  **Akihiko Torii**[*]

**Tomas Pajdla**[‡]  **Josef Sivic**[†,‡]

[†]Inria  [‡]CIIRC, CTU in Prague  [§]DeepMind  [*]Tokyo Institute of Technology

## Abstract

We address the problem of finding reliable dense correspondences between a pair of images. This is a challenging task due to strong appearance differences between the corresponding scene elements and ambiguities generated by repetitive patterns. The contributions of this work are threefold. First, inspired by the classic idea of disambiguating feature matches using semi-local constraints, we develop an end-to-end trainable convolutional neural network architecture that identifies sets of spatially consistent matches by analyzing neighbourhood consensus patterns in the 4D space of all possible correspondences between a pair of images without the need for a global geometric model. Second, we demonstrate that the model can be trained effectively from weak supervision in the form of matching and non-matching image pairs without the need for costly manual annotation of point to point correspondences. Third, we show the proposed neighbourhood consensus network can be applied to a range of matching tasks including both category- and instance-level matching, obtaining the state-of-the-art results on the PF Pascal dataset and the InLoc indoor visual localization benchmark.

## 1 Introduction

Finding visual correspondences is one of the fundamental image understanding problems with applications in 3D reconstruction [2], visual localization [32, 42] or object recognition [21]. In recent years, significant effort has gone into developing trainable image representations for finding correspondences between images under strong appearance changes caused by viewpoint or illumination variations [3, 4, 10, 13, 17, 37, 38, 44, 45]. However, unlike in other visual recognition tasks, such as image classification or object detection, where trainable image representations have become the *de facto* standard, the performance gains obtained by trainable features over the classic hand-crafted ones have been only modest at best [36].

One of the reasons for this plateauing performance could be the currently dominant approach for finding image correspondence based on matching *individual* image features. While we have now better local patch descriptors, the matching is still performed by variants of the nearest neighbour assignment in a feature space followed by separate disambiguation stages based on geometric constraints. This approach has, however, fundamental limitations. Imagine a scene with textureless regions or repetitive patterns, such as a corridor with almost textureless walls and only few distinguishing features. A small patch of an image, depicting a repetitive pattern or a textureless area, is indistinguishable from

---

[†]WILLOW project, Département d'informatique de l'École normale supérieure, ENS/INRIA/CNRS UMR 8548, PSL Research University, Paris, France.

[‡]CIIRC – Czech Institute of Informatics, Robotics and Cybernetics at the Czech Technical University in Prague, Czechia.

other portions of the image depicting the same repetitive or textureless pattern. Such matches will be either discarded [23] or incorrect. As a result, matching individual patch descriptors will often fail in such challenging situations.

In this work we take a different direction and develop a trainable neural network architecture that disambiguates such challenging situations by analyzing local neighbourhood patterns in a full set of dense correspondences. The intuition is the following: in order to disambiguate a match on a repetitive pattern, it is necessary to analyze a larger context of the scene that contains a unique non-repetitive feature. The information from this unique match can then be propagated to the neighbouring uncertain matches. In other words, the certain unique matches will *support* the close-by uncertain ambiguous matches in the image.

This powerful idea goes back to at least 1990s [5, 34, 35, 39, 47], and is typically known as *neighbourhood consensus* or more broadly as *semi-local constraints*. The neighbourhood consensus has been typically carried out on sparsely detected local invariant features as a filtering step performed *after* a hard assignment of features by nearest neighbour matching using the Euclidean distance in the feature space. Furthermore, the neighbourhood consensus has been evaluated by manually engineered criteria, such as a certain number of locally consistent matches [5, 34, 39], or consistency in geometric parameters including distances and angles between matches [35, 47].

In this work, we go one step further and propose a way of *learning* neighbourhood consensus constraints directly from training data. Moreover, we perform neighbourhood consensus *before* hard assignment of feature correspondence; that is, on the complete set of dense pair-wise matches. In this way, the decision on matching assignment is done only after taking into account the spatial consensus constraints, hence avoiding errors due to early matching decisions on ambiguous, repetitive or textureless matches.

**Contributions.** We present the following contributions. First, we develop a neighbourhood consensus network – a convolutional neural network architecture for dense matching that learns local geometric constraints between neighbouring correspondences without the need for a global geometric model. Second, we show that parameters of this network can be trained from scratch using a weakly supervised loss-function that requires supervision at the level of image pairs without the need for manually annotating individual correspondences. Finally, we show that the proposed model is applicable to a range of matching tasks producing high-quality dense correspondences, achieving state-of-the-art results on both category- and instance-level matching benchmarks. Both training code and models are available at [1].

## 2 Related work

This work relates to several lines of research, which we review below.

**Matching with hand-crafted image descriptors.** Traditionally, correspondences between images have been obtained by hand crafted local invariant feature detectors and descriptors [23, 25, 43] that were extracted from the image with a controlled degree of invariance to local geometric and photometric transformations. Candidate (tentative) correspondences were then obtained by variants of nearest neighbour matching. Strategies for removing ambiguous and non-distinctive matches include the widely used second nearest neighbour ratio test [23], or enforcing matches to be mutual nearest neighbours. Both approaches work well for many applications, but have the disadvantage of discarding many correct matches, which can be problematic for challenging scenes, such as indoor spaces considered in this work that include repetitive and textureless areas. While successful, handcrafted descriptors have only limited tolerance to large appearance changes beyond the built-in invariance.

**Matching with trainable descriptors.** The majority of trainable image descriptors are based on convolutional neural networks (CNNs) and typically operate on patches extracted using a feature detector such as DoG [23], yielding a sparse set of descriptors [3, 4, 10, 17, 37, 38] or use a pre-trained image-level CNN feature extractor [26, 33]. Others have recently developed trainable methods that comprise both feature detection and description [7, 26, 44]. The extracted descriptors are typically compared using the Euclidean distance, but an appropriate similarity score can be also learnt in a discriminative manner [13, 45], where a trainable model is used to both extract descriptors and

produce a similarity score. Finding matches consistent with a geometric model is typically performed in a separate post-processing stage [3, 4, 7, 10, 17, 22, 26, 37, 38, 44].

**Trainable image alignment.** Recently, end-to-end trainable methods have been developed to produce correspondences between images according to a parametric geometric model, such as an affine, perspective or thin-plate spline transformation [28, 29]. In these works, all pairwise feature matches are computed and used to estimate the geometric transformation parameters using a CNN. Unlike previous methods that capture only a sparse set of correspondences, this geometric estimation CNN captures interactions between a full set of dense correspondences. However, these methods currently only estimate a low complexity parametric transformation, and therefore their application is limited to only coarse image alignment tasks. In contrast, we target a more general problem of identifying reliable correspondences between images of a general 3D scene. Our approach is not limited to a low dimensional parametric model, but outputs a generic set of locally consistent image correspondences, applicable to a wide range of computer vision problems ranging from category-level image alignment to camera pose estimation. The proposed method builds on the classical ideas of neighbourhood consensus, which we review next.

**Match filtering by neighbourhood consensus.** Several strategies have been introduced to decide whether a match is correct or not, given the supporting evidence from the neighbouring matches. The early examples analyzed the patterns of distances [47] or angles [35] between neighbouring matches. Later work simply counts the number of consistent matches in a certain image neighbourhood [34, 39], which can be built in a scale invariant manner [31] or using a regular image grid [5]. While simple, these techniques have been remarkably effective in removing random incorrect matches and disambiguating local repetitive patterns [31]. Inspired by this simple yet powerful idea we develop a neighbourhood consensus network – a convolutional neural architecture that (i) analyzes the *full set of dense matches* between a pair of images and (ii) *learns* patterns of locally consistent correspondences directly from data.

**Flow and disparity estimation.** Related are also methods that estimate optical flow or stereo disparity such as [6, 15, 16, 24, 40], or their trainable counterparts [8, 19, 41]. These works also aim at establishing reliable point to point correspondences between images. However, we address a more general matching problem where images can have large viewpoint changes (indoor localization) or major changes in appearance (category-level matching). This is different from optical flow where image pairs are usually consecutive video frames with small viewpoint or appearance changes, and stereo where matching is often reduced to a local search around epipolar lines. The optical flow and stereo problems are well addressed by specialized methods that explicitly exploit the problem constraints (such as epipolar line constraint, small motion, smoothness, etc.).

## 3 Proposed approach

In this work, we combine the robustness of neighbourhood consensus filtering with the power of trainable neural architectures. We design a model which learns to discriminate a reliable match by recognizing patterns of supporting matches in its neighbourhood. Furthermore, we do this in a fully differentiable way, such that this trainable matching module can be directly combined with strong CNN image descriptors. The resulting pipeline can then be trained in an end-to-end manner for the task of feature matching. An overview of our proposed approach is presented in Fig. 1. There are five main components: (i) dense feature extraction and matching, (ii) the neighbourhood consensus network, (iii) a soft mutual nearest neighbour filtering, (iv) extraction of correspondences from the output 4D filtered match tensor, and (v) weakly supervised training loss. These components are described next.

### 3.1 Dense feature extraction and matching

In order to produce an end-to-end trainable model, we follow the common practice of using a deep convolutional neural network (CNN) as a dense feature extractor.

Then, given an image $I$, this feature extractor will produce a dense set of descriptors, $\{f_{ij}^I\} \in \mathbb{R}^d$, with indices $i = 1, \ldots, h$ and $j = 1, \ldots, w$, and $(h, w)$ denoting the number of features along image height and width (*i.e.* the spatial resolution of the features), and $d$ the dimensionality of the features.

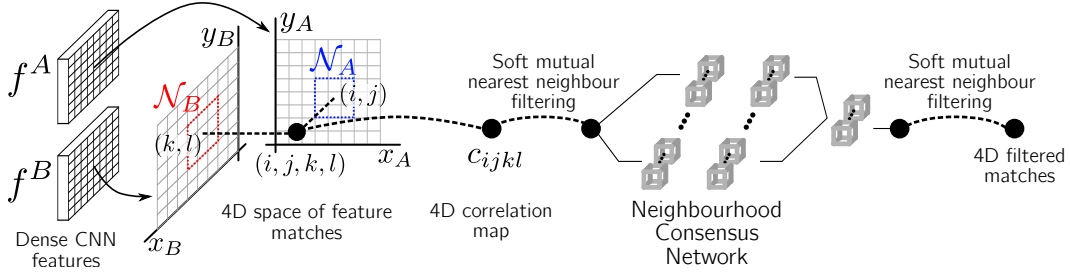

Figure 1: **Overview of the proposed method.** A fully convolutional neural network is used to extract dense image descriptors $f^A$ and $f^B$ for images $I_A$ and $I_B$, respectively. All pairs of individual feature matches $f^A_{ij}$ and $f^B_{kl}$ are represented in the 4-D space of matches $(i, j, k, l)$ (here shown as a 3-D perspective for illustration), and their matching scores stored in the 4-D correlation tensor $c$. These matches are further processed by the proposed soft-nearest neighbour filtering and neighbourhood consensus network (see Figure 2) to produce the final set of output correspondences.

While classic hand-crafted neighbourhood consensus approaches are applied *after* a hard assignment of matches is done, this is not well suited for developing a matching method that is differentiable and amenable for end-to-end training. The reason is that the step of selecting a particular match is not differentiable with respect to the set of all the possible features. In addition, in case of repetitive features, assigning the match to the first nearest neighbour might result in an incorrect match, in which case the hard assignment would lose valuable information about the subsequent closest neighbours.

Therefore, in order to have an approach that is amenable to end-to-end training, all pairwise feature matches need to be computed and stored. For this we use an approach similar to [28]. Given two sets of dense feature descriptors $f^A = \{f^A_{ij}\}$ and $f^B = \{f^B_{ij}\}$ corresponding to the images to be matched, the exhaustive pairwise cosine similarities between them are computed and stored in a 4-D tensor $c \in \mathbb{R}^{h \times w \times h \times w}$ referred to as *correlation map*, where:

$$c_{ijkl} = \frac{\langle f^A_{ij}, f^B_{kl} \rangle}{\|f^A_{ij}\|_2 \|f^B_{kl}\|_2}. \tag{1}$$

Note that, by construction, elements of $c$ in the vicinity of index $ijkl$ correspond to matches between features that are in the local neighbourhoods $\mathcal{N}_A$ and $\mathcal{N}_B$ of descriptors $f^A_{ij}$ in image $A$ and $f^B_{kl}$ in image $B$, respectively, as illustrated in Fig. 1; this structure of the 4-D correlation map tensor $c$ will be exploited in the next section.

## 3.2 Neighbourhood consensus network

The correlation map contains the scores of *all* pairwise matches. In order to further process and filter the matches, we propose to use 4-D convolutional neural network (CNN) for the neighbourhood consensus task (denoted by $N(\cdot)$), which is illustrated in Fig. 2.

Determining the correct matches from the correlation map is, *a priori*, a significant challenge. Note that the number of correct matches are of order of $hw$, while the size of the correlation map is of the order of $(hw)^2$. This means that the great majority of the information in the correlation map corresponds to *matching noise* due to incorrectly matched features.

However, supported by the idea of neighbourhood consensus presented in Sec. 1, we can expect correct matches to have a coherent set of supporting matches surrounding them in the 4-D space. These geometric patterns are equivariant with translations in the input images; that is, if the images are translated, the matching pattern is also translated in the 4-D space by an equal amount. This property motivates the use of 4-D convolutions for processing the correlation map as the same operations should be performed regardless of the location in the 4-D space. This is analogous to the motivation for using 2-D convolutions to process individual images – it makes sense to use convolutions, instead of for example a fully connected layer, in order to profit from weight sharing and keep the number of trainable parameters low. Furthermore, it facilitates sample-efficient training as a single training example provides many error signals to the convolutional weights, since the *same* weights are applied at all positions of the correlation map. Finally, by processing matches with a 4D convolutional network we establish a strong locality prior on the relationships between the matches. That is, by

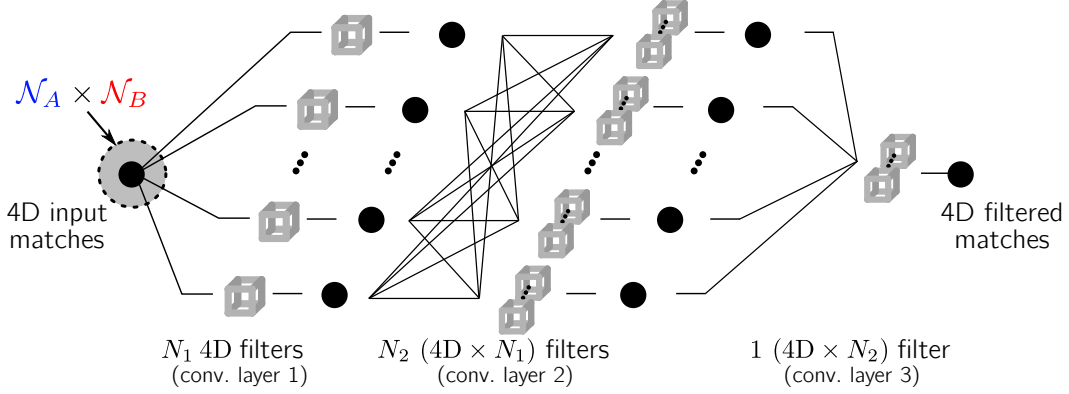

4D input matches

$\mathcal{N}_A \times \mathcal{N}_B$

4D filtered matches

$N_1$ 4D filters
(conv. layer 1)

$N_2$ (4D $\times$ $N_1$) filters
(conv. layer 2)

1 (4D $\times$ $N_2$) filter
(conv. layer 3)

Figure 2: **Neighbourhood Consensus Network (NC-Net).** A neighbourhood consensus CNN operates on the 4D space of feature matches. The first 4D convolutional layer filters span $\mathcal{N}_A \times \mathcal{N}_B$, the Cartesian product of local neighbourhoods $\mathcal{N}_A$ and $\mathcal{N}_B$ in images $A$ and $B$ respectively. The proposed 4D neighbourhood consensus CNN can learn to identify the matching patterns of reliable and unreliable matches, and filter the matches accordingly.

design, the network will determine the quality of a match by examining only the information in a local 2D neighbourhood in each of the two images.

The proposed neighbourhood consensus network has several convolutional layers, as illustrated in Fig. 2, each followed by ReLU non-linearities. The convolutional filters of the first layer of the proposed CNN span a local 4-D region of the matches space, which corresponds to the Cartesian product of local neighbourhoods $\mathcal{N}_A$ and $\mathcal{N}_B$ in each image, respectively. Therefore, each 4-D filter of the first layer can process and detect patterns in all pairwise matches of these two neighbourhoods. This first layer has $N_1$ filters that can specialize in learning different local geometric deformations, producing $N_1$ output channels, that correspond to the agreement with these local deformations at each 4-D point of the correlation tensor. These output channels are further processed by subsequent 4-D convolutional layers. The aim is that these layers capture more complex patterns by combining the outputs from the previous layer, analogously to what has been observed for 2-D CNNs [46]. Finally, the neighbourhood consensus CNN produces a single channel output, which has the same dimensions as the 4D input matches.

Finally, in order to produce a method that is invariant to the particular order of the input images, that is, that it will produce the same matches regardless of whether an image pair is input to the net as $(I^A, I^B)$ or $(I^B, I^A)$, we propose to apply the network twice in the following way:

$$\tilde{c} = N(c) + \left( N(c^{\mathrm{T}}) \right)^{\mathrm{T}}, \tag{2}$$

where by $c^{\mathrm{T}}$ we mean swapping the pair of dimensions corresponding to the first and second images: $\left( c^{\mathrm{T}} \right)_{ijkl} = c_{klij}$. This final output constitutes the *filtered matches* $\tilde{c}$ using the neighbourhood consensus network, where matches with inconsistent *local* patterns are downweighted or removed. Further filtering can be done by means of a *global* filtering strategy, as presented next.

### 3.3 Soft mutual nearest neighbour filtering

Although the proposed neighbourhood consensus network can suppress and amplify matches based on the supporting evidence in their neighbourhoods – that is, at a semi-local level – it cannot enforce global constraints on matches, such as being a *reciprocal* match, where matched features are required to be mutual nearest neighbours:

$$(f_{ab}^A, f_{cd}^B) \text{ mutual N.N.} \iff \begin{cases} (a,b) = \arg\min_{ij} \|f_{ij}^A - f_{cd}^B\| \\ (c,d) = \arg\min_{kl} \|f_{ab}^A - f_{kl}^B\|. \end{cases} \tag{3}$$

Filtering the matches by imposing the hard mutual nearest neighbour condition expressed by (3) would eliminate the great majority of candidate matches, which makes it unsuitable for usage in an end-to-end trainable approach, as this hard decision is non-differentiable.

We therefore propose a softer version of the mutual nearest neighbour filtering ($M(\cdot)$), both in the sense of *softer decision* and *better differentiability properties*, that can be applied on dense 4-D match scores:

$$\hat{c} = M(c), \quad \text{where} \quad \hat{c}_{ijkl} = r^A_{ijkl} r^B_{ijkl} c_{ijkl}, \tag{4}$$

and $r^A_{ijkl}$ and $r^B_{ijkl}$ are the ratios of the score of the particular match $c_{ijkl}$ with the best scores along each pair of dimensions corresponding to images $A$ and $B$ respectively:

$$r^A_{ijkl} = \frac{c_{ijkl}}{\max_{ab} c_{abkl}}, \quad \text{and} \quad r^B_{ijkl} = \frac{c_{ijkl}}{\max_{cd} c_{ijcd}}. \tag{5}$$

This soft mutual nearest neighbour filtering operates as a gating mechanism on the input, downweighting the scores of matches that are not mutual nearest neighbours. Note that the proposed formulation is indeed a *softer* version of the mutual nearest neighbours criterion as $\hat{c}_{ijkl}$ equals the matching score $c_{ijkl}$ iff $(f^A_{ij}, f^B_{kl})$ are mutual nearest neighbours, and is decreased to a value in $[0, c_{ijkl})$ otherwise. On the contrary, the "hard" mutual nearest neighbour matching would assign $\hat{c}_{ijkl} = 0$ in the latter case.

While this filtering step has no trainable parameters, it can be inserted in the CNN pipeline at both training and evaluation stages, and it will help to enforce the global *reciprocity* constraint on matches. In the proposed approach, the soft mutual nearest neighbour filtering is used to filter both the correlation map, as well as the output of the neighbourhood consensus CNN, as illustrated in Fig. 1.

### 3.4 Extracting correspondences from the correlation map

Suppose that we want to match two images $I^A$ and $I^B$. Then, the output of our model will produce a 4-D filtered correlation map $c$, which contains (filtered) scores for all pairwise matches. However, for various applications, such as image warping, geometric transformation estimation, pose estimation, visualization, etc, it is desirable to obtain a set of point-to-point image correspondences between the two images. To achieve this, a hard assignment can be performed in either of two possible directions, from features in image $A$ to features in image $B$, or vice versa.

For this purpose, two scores are defined from the correlation map, by performing soft-max in the dimensions corresponding to images $A$ and $B$:

$$s^A_{ijkl} = \frac{\exp(c_{ijkl})}{\sum_{ab} \exp(c_{abkl})} \quad \text{and} \quad s^B_{ijkl} = \frac{\exp(c_{ijkl})}{\sum_{cd} \exp(c_{ijcd})}. \tag{6}$$

Note that the scores are: (i) positive, (ii) normalized using the soft-max function, which makes $\sum_{ab} s^B_{ijab} = 1$. Hence we can interpret them as discrete conditional probability distributions of $f^A_{ij}, f^B_{kl}$ being a match, given the position $(i, j)$ of the match in $A$ or $(k, l)$ in $B$. If we denote $(I, J, K, L)$ the discrete random variables indicating the position of a match (*a priori* unknown), and $(i, j, k, l)$ the particular position of a match, then:

$$\mathbb{P}\left(K = k, L = l \mid I = i, J = j\right) = s^B_{ijkl} \quad \text{and} \quad \mathbb{P}\left(I = i, J = j \mid K = k, L = l\right) = s^A_{ijkl}. \tag{7}$$

Then, the hard-assignment in one direction can be done by just taking the most likely match (the mode of the distribution):

$$\begin{aligned}
f^B_{kl} \text{ assigned to a given } f^A_{ij} \iff (k, l) &= \arg\max_{cd} \mathbb{P}\left(K = c, L = d \mid I = i, J = j\right) \\
&= \arg\max_{cd} s^B_{ijcd},
\end{aligned} \tag{8}$$

and analogously to obtain the matches $f^A_{ij}$ assigned to a given $f^B_{kl}$.

This probabilistic intuition allows us to model the match uncertainty using a probability distribution and will be also useful to motivate the loss used for weakly-supervised training, which will be described next.

### 3.5 Weakly-supervised training

In this section we define the loss function used to train our network. One option is to use a strongly-supervised loss, but this requires dense annotations consisting of all pairs of corresponding points

for each training image pair. Obtaining such exhaustive ground-truth is complicated – dense manual annotation is impractical, while sparse annotation followed by an automatic densification technique typically results in imprecise and erroneous training data. Another alternative is to resort to synthetic imagery which would provide point correspondences by construction, but this has the downside of making it harder to generalize to larger appearance variations encountered in real image pairs we wish to handle. Therefore, it is desirable to be able to train directly from pairs of real images, and using as little annotation as possible.

For this we propose to use a training loss that only requires a weak-level of supervision consisting of annotation on the level of image pairs. These training pairs $(I^A, I^B)$ can be of two types, positive pairs, labelled with $y = +1$, or negative pairs, labelled with $y = -1$. Then, the following loss function is proposed:

$$\mathcal{L}(I^A, I^B) = -y\left(\bar{s}^A + \bar{s}^B\right),$$ (9)

where $\bar{s}^A$ and $\bar{s}^B$ are the mean matching scores over all hard assigned matches as per (8) of a given image pair $(I^A, I^B)$ in both matching directions.

Note that the minimization of this loss maximizes the scores of positive and minimizes the scores of negative image pairs, respectively. As explained in 3.4, the hard-assigned matches correspond to the modes of the distributions of (7). Therefore, maximizing the score forces the distribution towards a Kronecker delta distribution, having the desirable effect of producing well-identified matches in positive image pairs. Similarly, minimizing the score forces the distribution towards the uniform one, weakening the matches in the negative image pairs. Note that while the only scores that directly contribute to the loss are the ones coming from hard-assigned matches, all matching scores affect the loss because of the normalization in (6). Therefore, all matching scores will be updated at each training step.

## 4  Experimental results

The proposed approach was evaluated on both instance- and category-level matching problems. The same approach is used to obtain reliable matches for both problems, which are then used to solve two completely different tasks: (i) camera pose estimation in the challenging scenario of indoor localization, in the instance-level matching case, and (ii) semantic object alignment in the category-level matching case. Next we present the implementation details, followed by the results on the two tasks.

**Implementation details.**  The model was implemented in PyTorch [27], and a ResNet-101 network [14] initialized on ImageNet was used for feature extraction (up to the conv4_23 layer). The neighbourhood consensus network $N(\cdot)$ contains three layers of $5 \times 5 \times 5 \times 5$ filters or two layers of $3 \times 3 \times 3 \times 3$ filters for category level and instance level matching, respectively. In both cases, the intermediate results have 16 channels ($N_1 = N_2 = 16$). A feature resolution of $25 \times 25$ was used for training. As accurately localized features are needed for the pose estimation task, we extract correspondence for pose estimation at test time at a higher resolution resulting in a $200 \times 150$ feature map, which is downsampled using 4-D max+argmax pooling operation after computing the correlation map to $100 \times 75$ for increased efficiency. The model is initially trained for 5 epochs using Adam optimizer [20], with a learning rate of $5 \times 10^{-4}$ and keeping the feature extraction layer weights fixed. For category level matching, the model is then subsequently finetuned for 5 more epochs, training both the feature extraction and the neighbourhood consensus network, with a learning rate of $1 \times 10^{-5}$. In the case of instance level matching, finetuning the feature extraction did not improve the performance. Additional implementation details are given in the supplementary material [30].

### 4.1  Category-level matching

The proposed method was evaluated on the task of category-level matching, where, given two images containing different instances from the same category (*e.g.* two different cat images) the goal is to match similar semantic parts.

**Dataset and evaluation measure.**  We report results on the PF-Pascal [11] dataset, which contains 1,351 semantically related image pairs from the 20 object categories of the PASCAL VOC [9] dataset.

| Method | PCK |
|---|---|
| HOG+PF-LOM [11] | 62.5 |
| SCNet-AG+ [12] | 72.2 |
| CNNGeo [28] | 71.9 |
| WeakAlign [29] | 75.8 |
| **NC-Net** | **78.9** |

| Dist. (m) | SparsePE [42] | DensePE [42] | DensePE + MNN | DensePE + NC-Net | InLoc [42] | InLoc + MNN | InLoc + NC-Net |
|---|---|---|---|---|---|---|---|
| 0.25 | 21.3 | 35.3 | 31.9 | 37.1 | 38.9 | 37.1 | **44.1** |
| 0.50 | 30.7 | 47.4 | 50.5 | 53.5 | 56.5 | 60.2 | **63.8** |
| 1.00 | 42.6 | 57.1 | 62.0 | 62.9 | 69.9 | 72.0 | **76.0** |
| 2.00 | 48.3 | 61.1 | 64.7 | 66.3 | 74.2 | 76.3 | **78.4** |

Table 1: **Results for semantic keypoint transfer.** We show the rate (%) of correctly transferred keypoints within thresh. $\alpha = 0.1$.

Table 2: **Comparison of indoor localization methods.** We show the rate (%) of correctly localized queries within a given distance (m) and $10°$ angular error.

We follow the same evaluation protocol as [12, 29], and use the split from [12] which divides the dataset into approximately 700 pairs for training, 300 for validation and 300 for testing. In order to train the network in a weakly-supervised manner using the proposed loss (9), the 700 training pairs are used as positive training pairs, and negative pairs are generated by randomly pairing images of different categories, such as a car with a dog image. The performance is measured using the percentage of correct keypoints (PCK), that is, number of correctly matched annotated keypoints.

**Results.** Quantitative results are presented in Table 1. The proposed neighbourhood consensus network (NC-Net) obtains ~3% improvement over the state-of-the-art methods on this dataset [29]. An example of semantic keypoint transfer is shown in Figure 3 and demonstrates how our approach can correctly match semantic object parts in challenging situations with large changes of appearance and non-rigid geometric deformations. See the supplementary material [30] for additional examples.

## 4.2 Instance-level matching

Next we show that our method is also suitable for instance level matching and consider specifically the application to indoor visual localization, where the goal is to estimate an accurate 6DoF camera pose of a query photograph given a large-scale 3D model of a building. This is an extremely challenging instance-level matching task as indoor spaces are often self-similar and contain large textureless areas. We compare our method with the recently introduced indoor localization approach of [42], which is a strong baseline that outperforms several state-of-the-art methods, and introduces a challenging dataset for large scale indoor localization.

**Dataset and evaluation measure.** We evaluate on the InLoc dataset [42], which consists of 10K database images (perspective cutouts) extracted from 227 RGBD panoramas, and an additional set of 356 query images captured with a smart-phone camera at a different time from the database images. We follow the same evaluation protocol and report the percentage of correctly localized queries at a given camera position error threshold. As the InLoc dataset was designed for evaluation and does not contain a training set, we collected an Indoor Venues Dataset, consisting of user-uploaded photos, captured at public places such as restaurants, cafes, museums or cathedrals, by crawling Google Maps. It features similar appearance variations as the InLoc dataset, such as illumination changes, and scene modifications due to the passage of time. This dataset contains 3861 positive image pairs from 89 different venues in 6 different cities, split into *train*: 3481 pairs (80 places) and *validation*: 380 pairs (from the remaining 9 places). The design and collection procedures are described in the supplementary material [30] and the dataset is available at [1]. As in the case of category-level matching, negative pairs were generated by randomly sampling images from different places.

**Results.** We plug-in our trainable neighbourhood consensus network (NC-Net) as a correspondence module into the InLoc indoor localization pipeline [42]. We evaluate two variants of the approach. In the first variant, denoted DensePE+NC-Net, the DensePE [42] method is used for generating candidate image pairs, and then our network (NC-Net) is used to produce the correspondences that are employed for pose estimation. In the second variant, denoted InLoc+NC-Net, we use the full InLoc pipeline, including pose-verification by view synthesis. In this case, matches produced by NC-Net are used as input for pose estimation for each of the top $N = 10$ candidate pairs from DensePE, and the resulting candidate poses are re-ranked using pose-verification by view-synthesis. As an ablation study, these two experiments are also performed when NC-Net is replaced with hard mutual

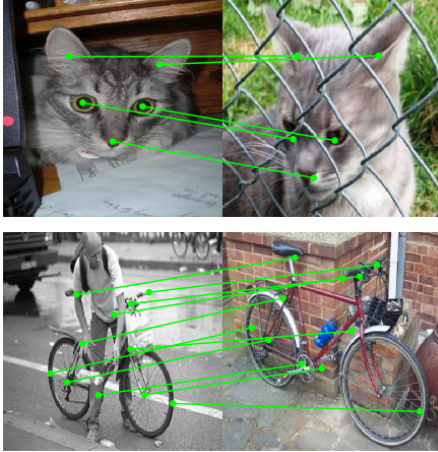

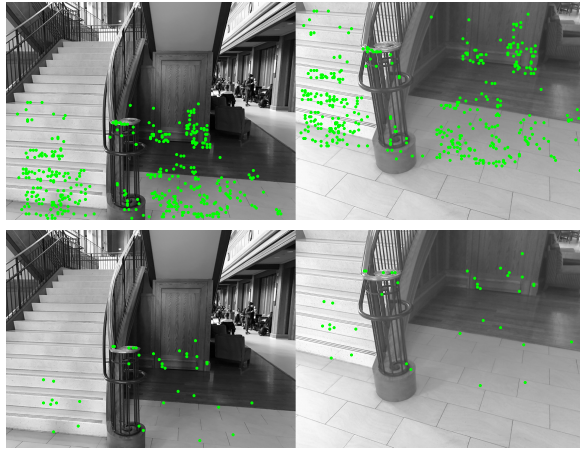

Figure 3: **Semantic keypoint transfer.** The annotated (ground truth) keypoints in the left image are automatically transferred to the right image using the dense correspondences between the two images obtained from our NC-Net.

Figure 4: **Instance-level matching.** Top row: inlier correspondences (shown as green dots) obtained by our approach (InLoc+NC-Net). Bottom row: Baseline inlier correspondences (InLoc+MNN). Our method provides a much larger and locally consistent set of matches, even in low-textured regions. Both methods use *the same* CNN features.

nearest neighbours matching (MNN), using the same base CNN network (ResNet-101). Results are summarised in Table 2 and clearly demonstrate benefits of our approach (DensePE+NC-Net) compared to both sparse keypoint (DoG+SIFT) matching (SparsePE) and the CNN feature matching used in [42] (DensePE). When inserted into the entire localization pipeline, our approach (InLoc + NC-Net) obtains state-of-the-art results on the indoor localization benchmark. An example of obtained correspondences on a challenging indoor scene with repetitive structures and texture-less areas is shown in figure 4. Additional results are shown in the supplementary material [30].

### 4.3 Limitations

While our method identifies correct matches in many challenging cases, some situations remain difficult. The two typical failure modes include: repetitive patterns combined with large changes in scale, and locally geometrically consistent groups of incorrect matches. Furthermore, the proposed method has quadratic $O(N^2)$ complexity with respect to the number of image pixels (or CNN features) $N$. This limits the resolution of the images that we are currently able to handle to $1600 \times 1200$px (or $3200 \times 2400$px if using the 4-D max+argmax pooling operation).

## 5  Conclusion

We have developed a neighbourhood consensus network — a CNN architecture that learns local patterns of correspondences for image matching without the need for a global geometric model. We have shown the model can be trained effectively from weak supervision and obtains strong results outperforming state-of-the-art on two very different matching tasks. These results open up the possibility for end-to-end learning of other challenging visual correspondence tasks, such as 3D category-level matching [18], or visual localization across day/night illumination [32].

### Acknowledgements

This work was partially supported by JSPS KAKENHI Grant Numbers 15H05313, 16KK0002, EU-H2020 project LADIO No. 731970, ERC grant LEAP No. 336845, CIFAR Learning in Machines & Brains program and the European Regional Development Fund under the project IMPACT (reg. no. CZ.02.1.01/0.0/0.0/15 003/0000468).

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
