[Reviews · NeurIPS 2018]

Reviewer 1



Summary: This paper propose a differentiable and an end-to-end approach to refine the correspondences (both semantic and instance level) by utilizing the neighborhood consensus on 4D correlation matrix (of feature map). The pipeline is evaluated for PASCAL keypoints dataset for semantic correspondences, and InLoc for instance level correspondences Pros: + intuitive, well described approach (I have some implementation level questions though, please see below) + clearly state of the art on previously used benchmarks. Cons: - Computational Time is linear with the increase in resolution. How to handle the scenarios where dense correspondences between a pair of HD images (1080x1920) is required? How much time does it take to compute dense correspondences between two images by the proposed approach? Is the compute time function of per pixel or it is a function of per image? - The approach is well motivated for dense correspondences but evaluation is done using sparse correspondences? How about using the datasets used for optical flow estimation (by approaches such as FlowNet or PWC-Net and many other papers)? Dense correspondences between a pair of images is equivalent for optical flow. Concerns/Implementation level details: 1. I am not certain as why soft mutual nearest neighbors are defined (L 185-190) this particular way? 2. What is the base network/features for the results in Table-1 for NC-Net (like [31] used in Table-2) OR is it directly done on raw pixels? 3. I am not sure how the disparity between the resolution at the training time (25 x 25) and test time (200 x 150) influence performance? Why do you need a low res feature descriptor for training? Is it because of the memory constraints provided by a GPU for training OR to include larger batch size? 4. The optical flow estimation approaches define a smoothness constraint (see Unflow in AAAI, 2017)? Can the neighborhood consensus ensures this smoothness automatically? OR is there way to explicitly encode in your formulation? 5. Will the code be publicly released? Suggestions: 1. L 180-181 describes the cyclic consistency. I would suggest authors to probably include reference to Zhou et al. CVPR 2016 and related work. 2. In my opinion, the idea of dense correspondences share a large similarity with work on optical flow estimation. As such, I would suggest to include the different works in computer vision literature on optical flow in related work (perhaps under trainable image alignment).

Reviewer 2



Paper overview: The authors propose a novel, end-to-end trainable deep architecture for finding dense correspondences between pairs of images. First, dense features are extracted from both images using a 2D CNN. Then, an exhaustive cross-correlation is performed between the feature maps to construct a 4D matching hyper-volume, which is fed to a "neighbourhood consensus network". This is, in essence, a 4D convolutional architecture which evaluates spatially local subsets of correspondences in order to detect latent consensus patterns, in a way reminiscent of traditional semi-local matching approaches. The final output of the network is a probability distribution over the set of dense matches between the images. The network is trained in a weakly-supervised manner, by processing pairs of matching / non matching images, without any explicit annotation on correspondences. Comments: The paper is very well written, clear and easy to follow. The authors manage to convincingly communicate the intuition motivating their approach, and the technical sections are easy to understand. The problem being tackled is a fundamental one, and the proposed approach is interesting and novel (but see question #2 below). The experimental evaluation is quite convincing, even if I would have liked to see some quantitative results on the accuracy of instance-level matching, in addition to the final localization results when the proposed method is used as a component of InLoc. Over all this is a very good submission, but there are two main points I would like the authors to clarify: 1) As far as I can understand, the weakly-supervised loss described in section 3.5 doesn't impose any kind of "physically meaningful" constraint on the matches. The network could learn to select _any_ set of correspondences, as long as the soft and hard assignments for the positive image pairs are consistent with each other, even if these correspondences do not make any sense given the content of the two images. In essence, what is being optimized here is close to an entropy loss over the matching probability distributions. Is this the case? Have you ever observed any "drifting" behavior during training that could be due to this? 2) The proposed approach shares some similarities with recent works on stereo matching such as [1], i.e. it works by processing a matching cost volume through a CNN. Can the authors comment on this? [1] Kendall, Alex, et al. "End-to-end learning of geometry and context for deep stereo regression." CoRR, vol. abs/1703.04309 (2017)

Reviewer 3



The paper proposes an end-to-end convolutional NN that produces candidate image correspondences. The paper proposes a network that uses 4D convolutions to identify consistent matches. The proposed method trains a model using a weak supervision in order to avoid exhaustive manual annotations. The paper presents experiments on semantic and image-feature correspondences where the proposed method shows improvement. Pros: - End-to-end convolutional NN architecture - Trained using weakly supervised methods Cons: - Insufficient experiments - Missing a discussion about limitations While I believe that the proposed network is an interesting idea, my main concern with this submission is that it lacks a few experiments and clarifications. First. In many matching applications, it is crucial to know the timings the matching procedure takes given that many CV-applications use exhaustive search. It would be interesting to know how long the net takes to establish candidate correspondences as a function of the number of features to match. Clearly, if this is faster than exhaustive search, then clearly this will become a very good contribution that deserves to be extended. In applications that estimate a geometric model (e.g., pose, homographies, essential matrix), it is also important to know about the number of false positives (FP) that this method generates. This is because these applications will use RANSAC, and the higher the number of FPs, the slower the convergence of RANSAC. Lastly, I think the paper is missing a section discussing the limitations of the proposed approach.